# Analysis of Spatiotemporal Evolution and Driving Forces of Vegetation from 2001 to 2020: A Case Study of Shandong Province, China

Dejin Dong [1],*, Ziliang Zhao [2], Hongdi Gao [3], Yufeng Zhou [2], Daohong Gong [4], Huaqiang Du [2] and Yuichiro Fujioka [5]

1  Graduate School of Integrated Sciences for Global Society, Kyushu University, Motooka 744, Nishi-ku, Fukuoka 819-0395, Japan
2  School of Environmental and Resources Science, Zhejiang A & F University, Hangzhou 311300, China; bright@stu.zafu.edu.cn (Z.Z.)
3  Zhejiang Forest Resources Monitoring Center, Hangzhou 310020, China
4  School of Geography and Environment, Jiangxi Normal University, Nanchang 330022, China
5  Faculty of Social and Cultural Studies, Kyushu University, Motooka 744, Nishi-ku, Fukuoka 819-0395, Japan
*  Correspondence: dong.dejin.983@s.kyushu-u.ac.jp

**Abstract:** As global climate change intensifies and human activities escalate, changes in vegetation cover, an important ecological indicator, hold significant implications for ecosystem protection and management. Shandong Province, a critical agricultural and economic zone in China, experiences vegetation changes that crucially affect regional climate regulation and biodiversity conservation. This study employed normalized difference vegetation index (NDVI) data, combined with climatic, topographic, and anthropogenic activity data, utilizing trend analysis methods, partial correlation analysis, and Geodetector to comprehensively analyze the spatiotemporal variations and primary driving factors of vegetation cover in Shandong Province from 2001 to 2020. The findings indicate an overall upward trend in vegetation cover, particularly in areas with concentrated human activities. Climatic factors, such as precipitation and temperature, exhibit a positive correlation with vegetation growth, while land use changes emerge as one of the key drivers influencing vegetation dynamics. Additionally, topography also impacts the spatial distribution of vegetation to a certain extent. This research provides a scientific basis for ecological protection and land management in Shandong Province and similar regions, supporting the formulation of effective vegetation restoration and ecological conservation strategies.

**Keywords:** climate change; geodetector; normalized difference vegetation index (NDVI); trend analysis; vegetation cover change





## 1. Introduction

Since the Anthropocene, particularly with the progression of the Industrial Revolution, human activities have had profound impacts on terrestrial ecosystems [1–3]. Vegetation, a crucial component of terrestrial ecosystems, not only participates in the carbon cycle through photosynthesis but also plays an essential role in regulating energy exchange [4]. It acts as a natural bridge between the lithosphere, atmosphere, and hydrosphere by influencing surface albedo and surface roughness [5,6]. In recent decades, the global climate has undergone unprecedented changes, significantly impacting vegetation growth and distribution [7]. With the widespread recognition of global warming, the scientific community has extensively studied the role of climate impacts on vegetation change as a central component of current environmental challenges [8–10]. Thus, understanding the impact of climate change on vegetation change, especially against the backdrop of continually changing human activities, is crucial for predicting future ecosystem dynamics and devising effective human intervention strategies. The dynamic evolution of vegetation

is a complex and prolonged process [11,12]. High-resolution long-term data are vital for exploring the spatiotemporal variations in vegetation cover. Satellite remote sensing technology, known for its accuracy, extensive coverage, continuity, and comprehensiveness, has been widely used in the fields of ecological conservation and climate change [13]. The normalized difference vegetation index (NDVI), derived from the difference between red and near-infrared reflectance relative to their sum, not only provides continuous long-term data but has also been extensively utilized to study the response of vegetation to environmental changes on seasonal, interannual, and interdecadal scales [14,15]. In recent years, research utilizing NDVI data has proliferated. Bhuyan et al. [16] compared the ring width index (RWI) time series of 69 forest sites worldwide with NDVI data at different time scales, discovering that the sum of NDVI in summer had the strongest explanatory power for RWI among all NDVI phenological indicators. Mao et al. [17] constructed monthly NDVI series in Northeast China from 1982 to 2009 using a pixel-by-pixel linear regression model. They found that, over the past 28 years, NDVI data at 95 meteorological stations were significantly correlated with monthly average temperature and precipitation. The spatial average value of summer NDVI exhibited a downward trend with rising temperatures and significantly decreased precipitation. Lv et al. [18] determined that temperature is the primary driving factor for NDVI changes and plays a key role in controlling NDVI accumulation, as evidenced by partial correlation analysis results of the distribution of NDVI and climatic factors on the Korean Peninsula.

The dynamics of vegetation and their driving factors have always been a focal point in ecological research, both domestically and internationally. Numerous studies have demonstrated that climatic factors such as precipitation and temperature significantly influence vegetation dynamics, with regional variations in these driving forces [19]. For instance, Jie Yang and colleagues explored the relationship between climate and vegetation dynamics along the Hu Line [20], while Yating Ren and others used Pearson correlation analysis to examine the relationship between vegetation dynamics and climatic factors in Jilin Province [21]. However, the processes of vegetation change are complex and influenced by a multitude of factors. Research also indicates that terrain, landforms, soil types, and changes in land use types have significant impacts on vegetation [22–25]. Despite these findings, the relative importance and interactions of these factors in the vegetation changes of plain and hilly regions remain unclear. Therefore, investigating the impacts of multiple driving factors on vegetation changes in these areas is a crucial research topic.

Additionally, traditional correlation and residual analysis methods that are commonly used to explore the drivers of vegetation change often fail to reveal the complex nonlinear relationships among multiple influencing factors, particularly the interactions between anthropogenic factors and climate fluctuations [26]. To overcome these inherent limitations, Geodetector [24], a nonlinear method, has been employed to elucidate the complex mechanisms driving NDVI changes. In recent years, Geodetector has gained prominence. Zhu et al. [27] employed the geographic detector method to quantify the impact of natural and human factors on NDVI changes and found that both types of factors drive NDVI variations. Land use type, annual average precipitation, and soil type were identified as having the greatest impact. Similarly, Zhang et al. [28] investigated the temporal and spatial changes and driving forces of NDVI in the Qinba Mountains of China from 1982 to 2015 using the geographic detector method. Their results indicated that the main factors affecting NDVI included rainfall, soil type, and elevation, while human activities (including population density) had a minimal impact on NDVI. Additionally, Yuan et al. [29] used geographic detectors to study the spatial heterogeneity of NDVI in the Heihe region of China. They found that precipitation was the primary factor influencing NDVI across the entire basin, with elevation and precipitation being the dominant factors in the upper and middle parts of the basin, respectively.

Located at the lower reaches of the Yellow River and adjacent to the Bohai and Yellow Seas, Shandong Province not only boasts rich vegetation resources [30] but also occupies a strategic position in China's economic landscape [31]. As an important agricultural produc-

tion area in China, coupled with rapid urbanization, Shandong's ecosystem is fragile and highly sensitive to climate change. Therefore, understanding the vegetation dynamics in Shandong Province is crucial for ecological protection, sustainable environmental development, agricultural productivity forecasting, and the formulation of policies and land use planning [32,33]. Although studies by Li et al. [34] and Yue et al. [35] have explored vegetation phenology and coverage changes in specific urban and delta regions, a comprehensive analysis of vegetation changes in Shandong Province is still lacking. Moreover, traditional linear methods such as those used by Shrestha et al. [36] often fall short in capturing the intricate, nonlinear interactions between multiple environmental factors and vegetation dynamics. Given these limitations, adopting advanced analytical methods like the Geodetector, as referenced in related literature [37–39] and considering regional characteristics, is vital. This method can detect and quantify these complex relationships, providing deeper insights into the driving factors behind vegetation changes in Shandong Province, which is essential for crafting wise environmental policies and enhancing agricultural resilience.

In this study, we employed a framework that integrated NDVI data with climatic, topographic, and anthropogenic activity data to examine the dynamics of vegetation changes in Shandong Province through trend analysis, partial correlation analysis, and Geodetector modeling. This study aims to analyze the spatio-temporal characteristics and trends of NDVI in the region from 2001 to 2020, explore the correlations between climatic factors and NDVI, and identify the key drivers affecting NDVI in Shandong. The results of this study will provide theoretical references for assessing the sustainable management and productivity of vegetation in Shandong.

## 2. Materials and Methods

### 2.1. General Situation of Study Area

Shandong Province is located in the eastern coastal region of China, in the lower reaches of the Yellow River (Figure 1a), with a total area of 154,300 square kilometers. The topography of Shandong is characterized by prominent central mountains, flat lowlands in the southwest and northwest, and gently rolling hills in the east. The central Tai–Lu–Yi Mountains serve as the provincial geographical center, with elevations gradually decreasing towards the periphery. The highest peak, Mount Tai, located in the central part of the province, reaches an elevation of 1518 m, while the lowest point lies within the Yellow River Delta in the north. The fundamental geomorphological types in Shandong include plains and mountainous hills (Figure 1c), with plains accounting for 55% of the area, primarily located in the northwestern and southwestern parts of the province. Mountainous and hilly areas constitute 29% of the terrain, predominantly found in the south–central and eastern regions. The predominant vegetation types are warm–temperate deciduous broadleaf forests, followed by coniferous forests and shrublands. Forested areas cover 26,100 square kilometers, mainly distributed in the south–central mountainous regions and the eastern low mountainous and hilly areas, with more scattered distributions in the plains. Shandong has a warm–temperate semi-humid monsoon climate, with an average annual temperature ranging from 11 to 14 °C, a frost-free period of 200–220 days, between 2400 and 2800 annual sunlight hours, and an average annual precipitation of 680 mm, which decreases from southeast to northwest. As of the end of 2020, the population of Shandong exceeded 103 million, predominantly engaged in agriculture and significantly influenced by human activities.

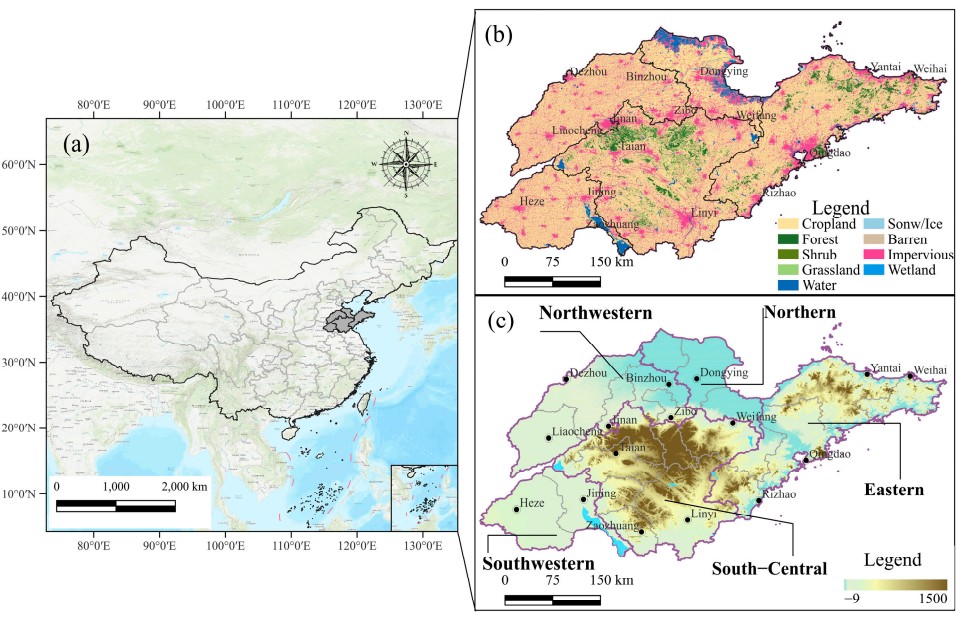

**Figure 1.** (**a**) Location of the study area in China, (**b**) land use of the study area, and (**c**) topography of the study area.

### 2.2. Data Sources and Preprocessing

All data used in this study and their sources are presented in Table 1. For investigating the drivers of vegetation change in Shandong Province, twelve potential driving factors were selected. These factors include precipitation, temperature, photosynthetically active radiation, elevation, slope, aspect, soil type, land use type, population density, nighttime light, distance to main rivers, and distance to roads. These influencing factors have been categorized into three groups: climatic factors, fundamental natural environmental factors, and human activities.

**Table 1.** Data description, source, and processing methods.

| Categories | Dataset | Abbreviation | Year Range and Resolution | Time Resolution | Data Source |
|---|---|---|---|---|---|
| / | Normalized difference vegetation index | NDVI | 2001–2020; 250 m | 16d | GEE |
| Climatic factors | Mean annual precipitation | PRE | 2001–2020; 1000 m | 1a | [40] |
| | Mean annual temperature | TEM | 2001–2020; 1000 m | 1a | [41] |
| | Photosynthetically active radiation | PAR | 2001–2020; 0.05° | 1a | GEE |
| Fundamental natural environmental factors | Elevation | Elevation | 2000; 30 m | | GEE |
| | Slope | Slope | 2000; 30 m | | GEE |
| | Aspect | Aspect | 2000; 30 m | | GEE |
| | Soil type | Soil | 2023; 1000 m | | Harmonized World Soil Database v2.0 |
| Human activities | Land use type | LAND | 2001–2020; 30 m | 1a | [42] |
| | Population density | POP | 2001–2020; 100 m | 1a | GEE |
| | Nighttime light | Light | 2001–2013; 1000 m | 1a | GEE |
| | | Light | 2014–2020; 750 m | 1a | GEE |
| | Distance to main rivers | River | 2020; 1000 m | | openstreetmap |
| | Distance to road | Road | 2020; 1000 m | | openstreetmap |

'16d' refers to a temporal resolution of 16 days, '1a' indicates an annual temporal resolution, and 'GEE' stands for Google Earth Engine, a cloud-based platform for earth observation and data analysis.

### 2.2.1. NDVI Dataset

The NDVI dataset is based on the Google Earth Engine cloud (https://earthengine.google.com/, accessed on 8 November 2023) computing platform, utilizing the MOD13Q1 V6.1 remote sensing imagery, which features a temporal resolution of 16 days and a spatial resolution of 250 m. To further mitigate the effects of cloud cover and aerosol scattering, the study selected qualified data products from the peak growing season (May to September) and generated 20-year NDVI datasets using the maximum value compositing method.

### 2.2.2. Climatic Factors

Temperature and precipitation datasets were sourced from the National Tibetan Plateau Scientific Data Center (http://data.tpdc.ac.cn, accessed on 8 November 2023), with a spatial resolution of 0.008333° (approximately 1 km) [40]. These datasets were downscaled in the region of China using the Delta spatial downscaling scheme, based on the global 0.5° climate dataset released by CRU and the global high-resolution climate dataset from WorldClim (https://www.worldclim.org/, accessed on 8 November 2023). The data include a 1 km resolution monthly precipitation dataset (0.1 mm) and a 1 km resolution monthly mean temperature dataset (0.1 °C). These datasets were generated for China using the Delta spatial downscaling scheme based on the global 0.5° climate dataset released by CRU and the global high-resolution climate dataset from WorldClim. The datasets were validated using data from 496 independent climate observation stations, ensuring their reliability. Using this dataset, the average annual temperature and annual cumulative precipitation for the study area were calculated. Photosynthetically active radiation (PAR) data were obtained from the MCD18C2 Collection 6.1 (GEE/061/MCD18C2) product, which provides daily PAR at 0.05° resolution [41]. The average PAR for each year was calculated from this dataset.

### 2.2.3. Fundamental Natural Environmental Factors

The digital elevation model (DEM) data, describing the terrain conditions, were sourced from Google Earth Engine (https://earthengine.google.com/, accessed on 8 November 2023). High-precision land use data were obtained from the China Land Cover Dataset (CLCD) (https://www.globallandcover.com, accessed on 8 November 2023), with a spatial resolution of 30 m. Slope, aspect, and elevation were derived from the DEM data.

### 2.2.4. Human Activities

Land use types were obtained from the China Land Cover Dataset (CLCD) (https://www.globallandcover.com, accessed on 8 November 2023), which provides high-precision insights with a spatial resolution of 30 m. Population density data were acquired from WorldPop (https://www.worldpop.org/, accessed on 8 November 2023), with a resolution of 100 m. Nighttime light data for 2001–2013 were sourced from the United States National Oceanic and Atmospheric Administration (NOAA) (GEE: NOAA/DMSP-OLS/NIGHTTIME_LIGHTS), and from 2014 to 2020 from NOAA (GEE: NOAA/VIIRS/DNB/MONTHLY_V1/VCMSLCFG). Basic road and river geographic information were obtained from OpenStreetMap (https://openstreetmap.org, accessed on 8 November 2023).

For the use of Geodetector, this research utilized QGIS 3.30 to create a fishnet tool, generating a grid of 2 km × 2 km across the entire study area, totaling 39,391 sampling points. Spatial attributes corresponding to X and Y values were extracted, and different influencing factors were categorized using natural break, geometric interval, and quantile methods, among others. Geographic detector analysis was then conducted using the GD package (version 10.3) in RStudio (version 2023.09.0 Build 463). All data were processed using the WGS 1984 geographic coordinate system. To ensure consistent resolution across the selected variables, bilinear resampling was employed, and the data were reprojected to a resolution of 1000 m.

### 2.3. Analysis of Methods

In this study, the dynamic characteristics, spatiotemporal evolution trends, driving factors, and the contributions of these drivers to vegetation cover in Shandong Province are examined. The research framework and specific tasks are illustrated in Figure 2. Utilizing the Google Earth Engine (GEE) platform and referencing pertinent literature, regional vegetation cover is characterized using the widely employed NDVI.

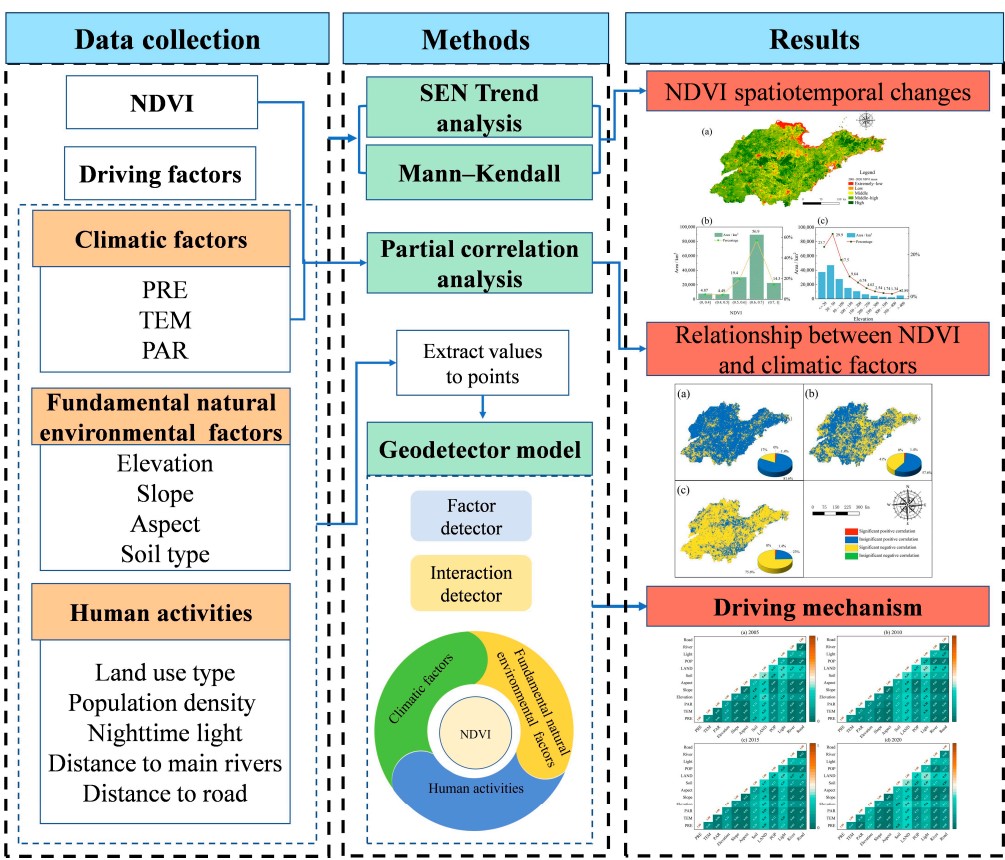

**Figure 2.** Brief technical flowchart of this study. 'NDVI' stands for normalized difference vegetation index, 'PRE' for precipitation, 'TEM' for temperature, 'PAR' for photosynthetically active radiation, and 'SEN' for the Theil–Sen median method.

### 2.3.1. Analysis of Vegetation Variation Trends

The Theil–Sen median method, also known as the Sen's slope estimator, is a non-parametric statistical technique used to calculate trends [43]. This method is favored for its computational efficiency and its insensitivity to outliers and measurement errors, making it particularly suitable for analyzing long-term time series of vegetation growth. In this study, the Theil–Sen median method combined with the Mann–Kendall trend test, both widely utilized in meteorology and hydrology, are employed to analyze the characteristics of NDVI changes in Shandong Province from 2001 to 2020. The formula for the Theil–Sen estimator is as follows:

$$\beta = Median\left(\frac{x_j - x_i}{j - i}\right),\ 2001 \leq i \leq j \leq 2020 \tag{1}$$

where $j$ and $i$ represent data points in the time series and $\beta$ represents Sen's slope. A positive value of $\beta$ suggests an upward trend in the series, while a negative value indicates a downward trend. A value close to zero suggests that changes in the time series are not significant.

The Mann–Kendall significance test, also known as the M-K test, is employed to assess the significance of trends over long-term time series data [44,45]. This method, along with the Sen's slope estimator, does not require the data to be normally distributed, thus making the results less susceptible to outliers. Consequently, this study utilizes the Mann–Kendall test to determine the significance of interannual changes in NDVI in Shandong Province.

The formula involves the following steps:

a.    Calculation of the $S$ statistic:

$$S = \sum_{i=1}^{n-1} \sum_{j=i+1}^{n} \text{sgn}(x_j - x_i) \tag{2}$$

$$\text{sgn}(x_j - x_i) = \begin{cases} 1, x_j - x_i > 0 \\ 0, x_j - x_i = 0 \\ -1, x_j - x_i < 0 \end{cases} \tag{3}$$

where $S$ is the Mann–Kendall test statistic, $n$ is the length of the time series, and $x_j$ and $x_i$ are sequential data values in time series $j$ and $i$. The statistic $S$ is nearly a normal distribution when sample sizes are larger than ten; the sgn denotes the sign function. The variance $Var(S)$ is calculated using Equation (3).

b.    Calculation of variance:

$$Var(S) = \frac{n(n-1)(2n+5)}{18} \tag{4}$$

c.    Calculation of the $Z$ statistic:

$$Z = \begin{cases} \frac{S-1}{\sqrt{\text{var}(S)}}, & S > 0 \\ 0, & S = 0 \\ \frac{S+1}{\sqrt{\text{var}(S)}}, & S < 0 \end{cases} \tag{5}$$

where $Z$ is the statistic normalized by the Mann–Kendall test and follows a normal distribution, and Var($S$) represents the variance. The test statistic $Z$ is used to test the trend. Maghsoodloo [46] proved that the statistic $S$ roughly follows a normal distribution when $n \geq 8$, and $Z$ is the standard normal distribution test statistic for $S$.

Combining the values of $\beta$ and $|Z|$, the trends in NDVI are classified into five categories. In this study, due to the practical absence of regions where $\beta$ is exactly zero, the interval ranging from $-0.001$ to $0.001$ was defined as stable and constant [47]. The outcomes of the Mann–Kendall test, conducted at a confidence level of 0.05, categorized the changes as either significant ($|Z| > 1.96$) or not significant ($|Z| \leq 1.96$) [48]. The method of distinguishing the significance of the trends is shown in Table 2.

**Table 2.** Types of change trends in NDVI based on Theil–Sen median method and Mann–Kendall test.

| $\beta$ | $\|Z\|$ | Trend Type | Trend Features |
|---|---|---|---|
| $\beta > 0.001$ | $\|Z\| > 1.96$ | 5 | Significant improvement |
| $\beta > 0.001$ | $\|Z\| \leq 1.96$ | 4 | Slight improvement |
| $\|\beta\| \leq 0.001$ | $\|Z\| \leq 1.96$ | 3 | Stable and unchanged |
| $\beta < -0.001$ | $\|Z\| \leq 1.96$ | 2 | Slight degradation |
| $\beta < -0.001$ | $\|Z\| > 1.96$ | 1 | Severe degradation |

$\beta$ represents Sen's slope. $Z$ is the statistic normalized by the Mann–Kendall test and follows a normal distribution.

### 2.3.2. Partial Correlation Analysis

To precisely control for the influences of confounding variables and clearly delineate the direct relationships between two variables, we employed partial correlation analysis. This approach avoids the potential pitfalls of regression analysis, which may obscure

indirect relationships through overfitting with multiple predictors, thus offering a more accurate understanding than simple linear correlation analysis [37]. Prior to conducting partial correlation analysis and in consideration of potential impacts from collinearity, we executed the Lilliefors test to assess the normal distribution and performed a collinearity analysis using the variance inflation factor (VIF) on the selected variables. The outcomes of these analyses indicated that the variables were free from influences of collinearity and distributional assumptions, thus validating the integrity of our statistical approach. The partial correlation coefficient (PCC) is employed to assess the degree of association between two variables, independent of the effects of other intervening variables. The formula for calculating the partial correlation coefficient is as follows:

$$R_{12,3} = \frac{r_{12} - r_{13}r_{23}}{\sqrt{\left(1 - r_{13}^2\right)\left(1 - r_{23}^2\right)}} \tag{6}$$

where $R_{12,3}$, $R_{13,2}$, and $R_{23,1}$ are the correlation coefficients among the variables; $R_{12,3}$ is the partial correlation coefficient between $r_1$ and $r_2$ after fixing the variable $r_3$. The value range of the partial correlation coefficient ranges from $-1$ to 1. When $R_{12,3} > 0$, the correlation is positive, meaning that both factors correlate in the same direction. When $R_{12,3} < 0$, the correlation is negative. The higher the partial correlation coefficient, the stronger the correlation between the two elements at the pixel.

The larger the PCC value is, the greater the effect of the variable on the NDVI. A smaller value indicates a weaker effect. The significance test for the partial correlation is shown in Formula (7), as follows:

$$t = \frac{r_{1\cdot23}}{\sqrt{1 - r_{1\cdot23}^2}}\sqrt{n - m - 1} \tag{7}$$

In this study, the partial correlations between the NDVI and climatic factors in Shandong were classified as significant positive correlations (PCC > 0, $p < 0.05$), nonsignificant positive correlations (PCC > 0, $p > 0.05$), significant negative correlations (PCC < 0, $p < 0.05$), and nonsignificant negative correlations (PCC < 0, $p > 0.05$).

### 2.3.3. Geographic Detector

Geodetector is a popular geostatistical model that analyzes spatial variations and reveals the driving factors behind them [24,49]. Geodetector consists of four subdetectors: factor detector, interaction detector, risk detector, and ecological detector. In this study, the former two detectors are used to investigate the driving mechanisms behind NDVI change [50,51].

(1)  Factor detector

The factor detector is calculated using the following q statistic:

$$q = 1 - \frac{\sum_{h=1}^{L} N_h \sigma_h^2}{N\sigma^2} = 1 - \frac{SSW}{SST} \tag{8}$$

where $0 \leq q \leq 1$, and the larger the value, the greater the explanatory power of the factor. When the $q$ value is 0, it means that the factor has no relationship with NDVI. $h$ is the number of strata for variables or factors, $N$ represents the number of units in stratum $h$, and $\sigma_h^2$ and $\sigma^2$ denote the variance in the stratum $h$ and the entire study area, respectively. $SSW$ and $SST$ denote the sum of squares within the data and the total sum of squares, respectively.

(2)  Interaction detector

The interaction detector identifies the interactions between different risk factors, specifically evaluating whether the combined effect of factors X1 and X2 increases or decreases the explanatory power for the dependent variable Y, or if their impacts on Y are independent. The

evaluation method involves calculating the *q*-values for each factor independently and then calculating the *q*-value for their interaction. The R software package is utilized to compute the interaction detector, which can be accessed at https://cran.r-project.org/web/packages/GD/, accessed on 8 November 2023.

## 3. Results

### 3.1. Temporal NDVI Analysis

Over the past two decades, the vegetation NDVI in Shandong Province has exhibited significant interannual variability, as illustrated in Figure 3. The provincial average annual NDVI value was 0.7252, indicating a relatively stable trend.

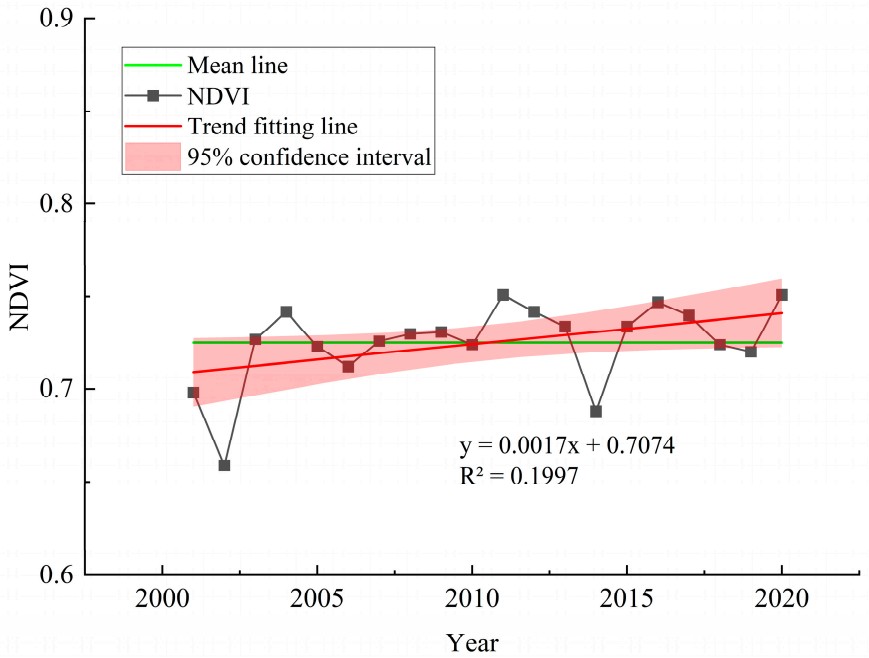

**Figure 3.** Annual average normalized difference vegetation index (NDVI) in Shandong from 2001 to 2020.

The linear regression analysis indicates a positive trend in the annual average NDVI values over the study period. The regression equation y = 0.0017x + 0.7074 suggests an annual increase in NDVI by 0.0017 units. The coefficient of determination ($R^2$ = 0.1997) indicates that approximately 20% of the variability in NDVI can be explained by the year, reflecting a modest fit of the model. Despite the relatively low $R^2$ value, the overall increase in NDVI suggests a slight improvement in vegetation coverage over time. The trend line, along with the 95% confidence interval, provides a visual representation of this increasing trend and the associated uncertainty.

Monthly data (Figure 4) reveal that NDVI values from January to May are generally lower, especially in January and February, where they range between 0.3 and 0.6. During the summer months of July and August, NDVI values significantly increase, exceeding 0.8, suggesting more vigorous vegetation growth in the warm season. In autumn and winter, from September to December, NDVI values gradually decrease but remain between 0.4 and 0.6, showing a relatively stable condition. The lowest annual average NDVI value occurred in 2002, at 0.659, while the highest values were recorded in 2011 and 2020, both at 0.751.

Figure 5 illustrates the sub-regional trends in NDVI changes in Shandong Province from 2001 to 2020. The linear regression coefficients for all four sub-regions are greater than 0, indicating a slow growth in vegetation over the 20-year period. The NDVI in the northwest region shows a higher average value of 0.806, indicating a relatively stable growth trend. In contrast, the northern region has an average NDVI of 0.5705, slightly below

the provincial average, with a more pronounced change in its slope, suggesting relatively unstable vegetation growth. The NDVI values in the eastern, south–cntral, and south-western regions fluctuate between 0.6 and 0.8. However, the interannual slope changes significantly between these areas, with the south–cntral region recording the highest slope at 0.0023 and the eastern region recording the lowest at 0.0002. The ecological resources in the south–cntral region, designated for soil conservation, water source conservation, and ecological restoration, are relatively stable, yet the increase in NDVI there is notably higher than in other areas. This may be closely associated with ecological restoration measures such as optimizing forest resources and enhancing water conservation.

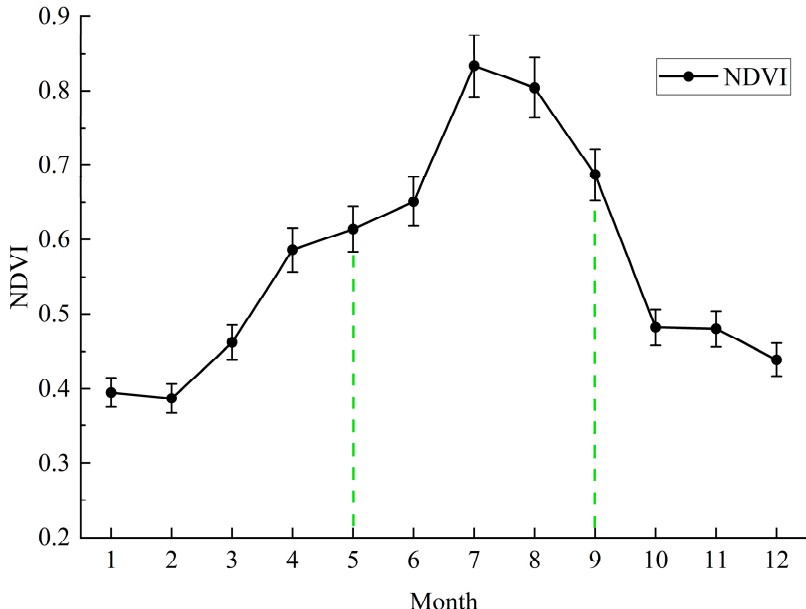

**Figure 4.** Monthly mean normalized difference vegetation index (NDVI) in Shandong from 2001 to 2020.

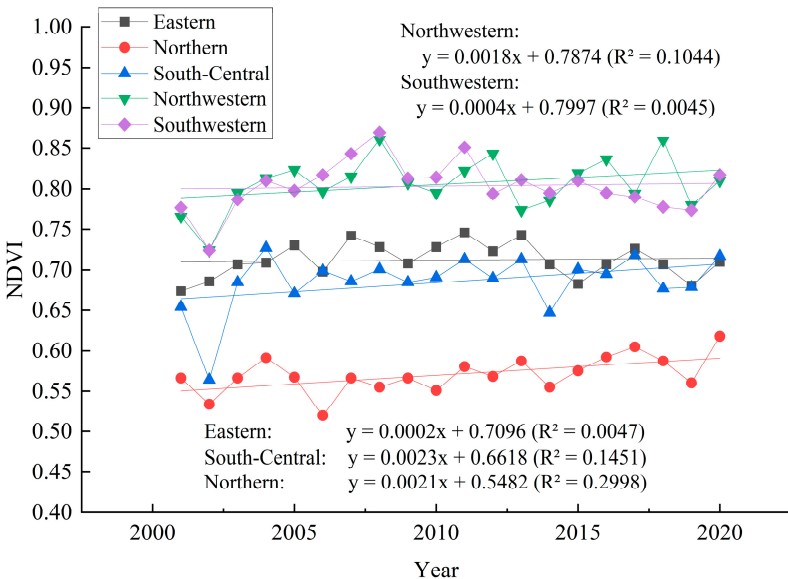

**Figure 5.** Changes in normalized difference vegetation index (NDVI) sub-regional trends in Shandong Province from 2001 to 2020.

### 3.2. Spatial Patterns of NDVI

Figure 6a,b show the spatial distribution of NDVI in Shandong Province from 2001 to 2020 and the area and proportion of NDVI distribution intervals, respectively. Figure 6a,b reveal the heterogeneity of ecological vegetation coverage across the province. Areas with NDVI values less than 0.5, constituting 9.36% of the total area, are primarily located in the northern, northwestern, and southwestern parts of Shandong. The northern region is characterized by coastal saline and alkaline soils, which support fewer plant species. The northwestern region, a major grain-producing area, is prone to soil desertification and salinization, making its vegetation growth sensitive. The southwestern region, also a significant grain-producing area, includes wetlands such as the Nan Si Lake and is a concentration area for mining subsidence. Areas with NDVI values greater than 0.7 account for 14.28% of the total area, mainly distributed in the northern parts of south–cntral and eastern Shandong. The northern part of south–cntral Shandong includes mountainous regions such as Mount Tai, Mount Lu, Mount Meng, and Mount Yi, while the northern part of eastern Shandong encompasses the Kunyu Mountain range. These areas, with varied topography and rich plant types, are the richest in forest resources and biodiversity within the province. They also serve as key zones for national and provincial conservation forests, playing crucial roles in water conservation and soil preservation.

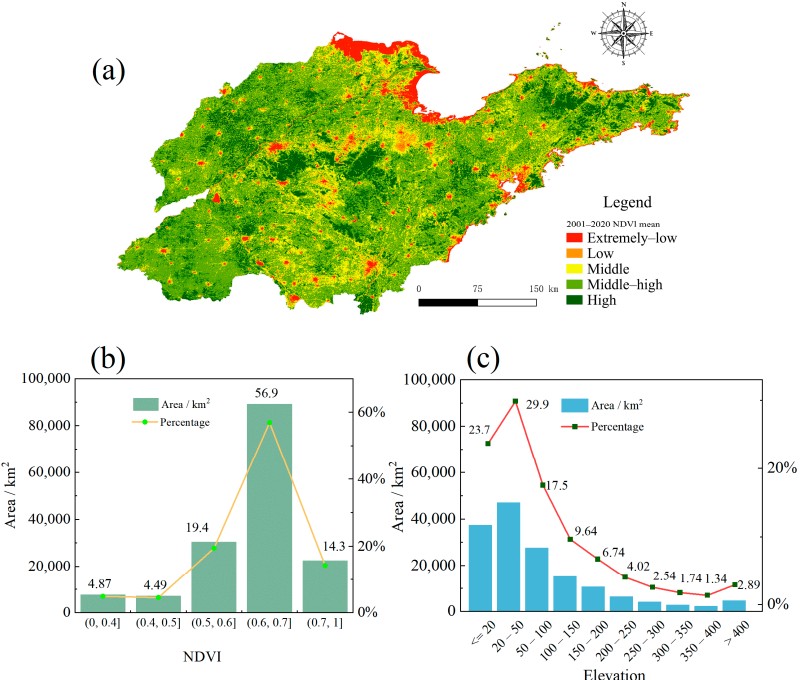

**Figure 6.** (**a**) Spatial distribution of average normalized difference vegetation index (NDVI) in Shandong Province, 2001–2020, (**b**) area and proportion of NDVI distribution intervals, and (**c**) area and proportion of NDVI by elevation range.

As shown in Figure 6c, NDVI initially decreases with increasing altitude, with the broadest distribution of NDVI occurring between elevations of 20 to 50 km, where vegetation is predominantly agricultural. Regions above 400 m constitute only 2.89% of the area. Overall, the distribution of NDVI in Shandong Province exhibits distinctive regional characteristics and ecological sensitivities.

Figure 7 shows the percentages of NDVI under different levels during 2001–2020. In this study, the natural breaks method is employed to classify annual NDVI into five categories: high coverage, middle–high coverage, middle coverage, low coverage, and extremely low coverage. Over the 20-year period, high-coverage areas (NDVI > 0.7) consistently dominated, accounting for the largest proportion of the total area each year. These

areas are predominantly located in the southern parts of Jinan, southern Zibo, northern Linyi, southern Dezhou, northern Yantai, and central Jining. The dominant land covers in these regions are forests and agricultural fields. This stable high vegetation coverage highlights the ecological vitality and the effectiveness of land management practices in these areas over the past two decades.

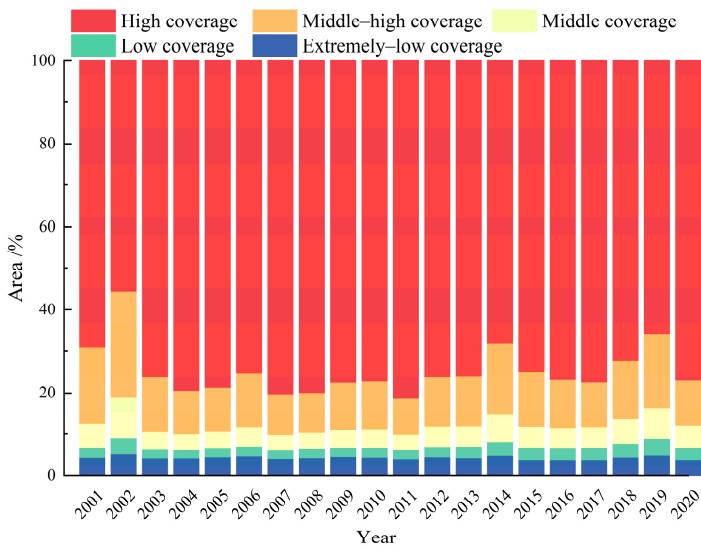

**Figure 7.** Percentages of normalized difference vegetation index (NDVI) under different levels during 2001–2020.

### 3.3. Analysis of NDVI Trends

Figures 8 and 9 show the NDVI trends across Shandong Province using the Theil–Sen slope method, validated by the Mann–Kendall trend test. We integrated the results of the Theil–Sen slope method and the Mann–Kendall trend test to classify NDVI trends into five categories, as presented in Table 2. Figure 8 shows the NDVI trends for four five-year intervals: 2001–2005, 2006–2010, 2011–2015, and 2016–2020. From 2001 to 2005, approximately 78% of the province showed slight improvement in NDVI, predominantly in the eastern and central regions, with only 16.3% exhibiting slight degradation, primarily in the northern areas. The average Sen's slope for the increasing trend was 0.047. During the 2006–2010 interval, 46.7% of the province continued to show slight improvement, especially in the central, eastern, and northern regions, whereas areas showing slight degradation rose to 44.9%, affecting mainly the coastal eastern and southwestern regions, with an average Sen's slope for the increasing trend of 0.038. Between 2011 and 2015, slight degradation was observed in 61% of the area, particularly in the northern and eastern regions, while 31.7% demonstrated slight improvement, mainly in the northwest and southern areas, with an average Sen's slope for the increasing trend of 0.043. In the final interval, 2016–2020, 52.5% of the province experienced slight degradation, especially in the northern regions, and 40% showed improvement, primarily in the eastern and central regions, with an average Sen's slope for the increasing trend of 0.039. Figure 9 shows the overall NDVI trend from 2000 to 2020, revealing significant improvement in 21.9% of the province, notably in the north, and slight improvement in 30.4%, particularly in the eastern and central regions. The average Sen's slope for the upward trend over the two decades was 0.008, signifying a steady increase in NDVI.

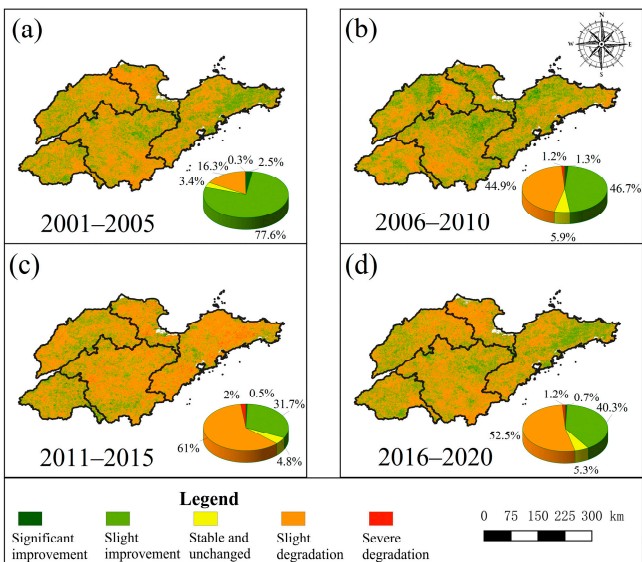

**Figure 8.** Interval distribution of normalized difference vegetation index (NDVI) change trends in Shandong Province. Green indicates significantly improved area, light green indicates slightly improved area, yellow indicates stable area, orange indicates slightly degraded area, and red indicates severely degraded area. The pie chart shows the proportion of different vegetation trend types.

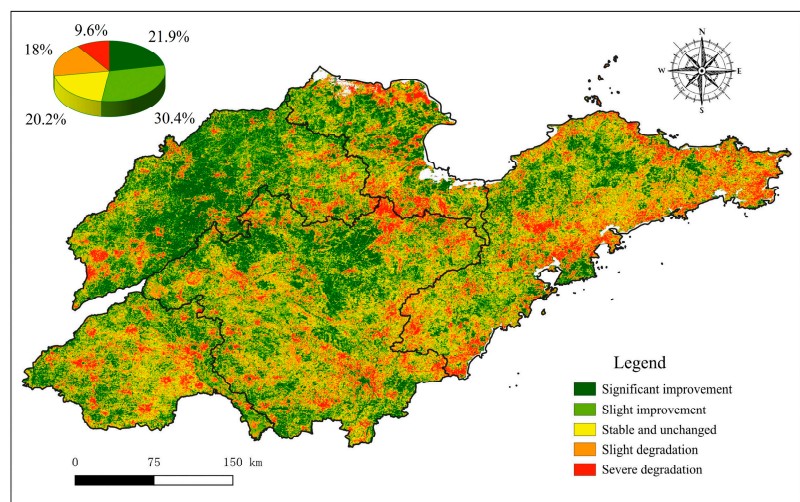

**Figure 9.** Distribution of normalized difference vegetation index (NDVI) change trends in Shandong Province from 2001 to 2020.

## 3.4. The Relationship between NDVI and Driving Factors

### 3.4.1. Spatiotemporal Response of NDVI to Climatic Factors in Shandong Province

Figures 10a and 11a demonstrate that the overall average PCC between NDVI and PRE is 0.203, indicating a significantly positive correlation in 1.4% of the area with no significant negative correlations observed. Figure 11a reveals that 83.0% of the region exhibits an overall positive correlation on average, while 17.0% shows negative correlations, predominantly located in the southwestern, central, and eastern parts of Shandong.

Figures 10b and 11b show that the overall average PCC between NDVI and TEM is 0.283, with 1.4% of the area showing significant positive correlations and no significant negative correlations noted. According to Figure 11b, 59.0% of the area exhibits an overall positive correlation on average, whereas 41.0% shows negative correlations, primarily in the eastern and south–cntral regions of Shandong.

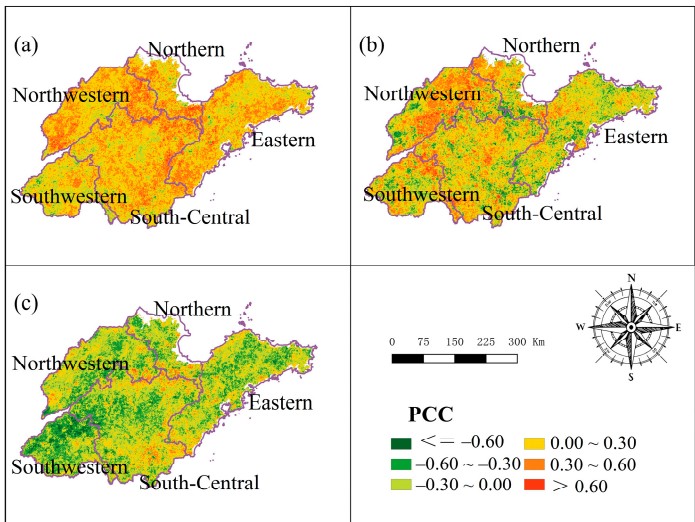

**Figure 10.** Partial correlation coefficients (PCCs) between the normalized vegetation index (NDVI) and annual precipitation (PRE) (**a**), annual mean temperature (TEM) (**b**), and photosynthetically active radiation (PAR) (**c**) in Shandong Province from 2001 to 2020.

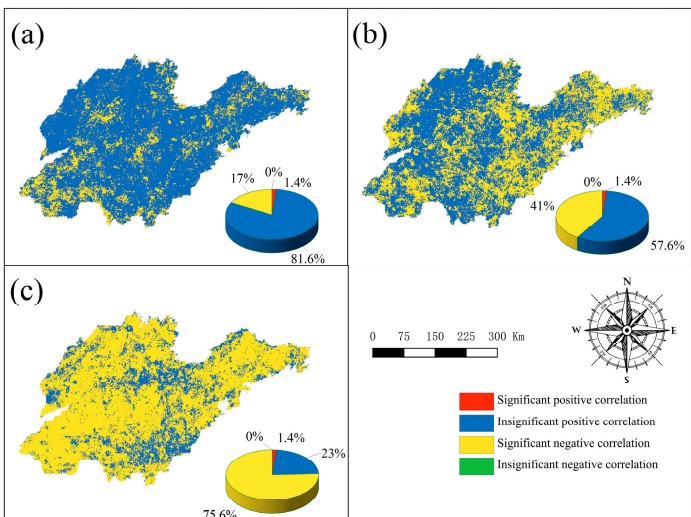

**Figure 11.** Sign and significance of partial correlations of normalized difference vegetation index (NDVI) with annual precipitation (PRE) (**a**), annual mean temperature (TEM) (**b**), and photosynthetically active radiation (PAR) (**c**) in Shandong Province from 2001 to 2020.

Figures 10c and 11c indicate that the overall average PCC between NDVI and PAR is 0.159, with 1.4% of the area exhibiting significant positive correlations and no significant negative correlations observed. Figure 11c shows that 24.4% of the area displays an overall positive correlation on average, while 75.6% presents negative correlations, with positive correlations mainly concentrated in the southern and central parts of Shandong.

### 3.4.2. The Relationship between Topography and NDVI in Shandong Province

As shown in Figure 12a, the highest NDVI values are observed on the northeast-facing slopes, averaging 0.628, followed closely by the northwest-facing slopes at 0.627, while the north-facing slopes exhibit the lowest NDVI, at 0.58. Overall, the NDVI is significantly higher on sunlit slopes compared to shaded slopes. Figure 12b illustrates that the majority of grid cells have a slope of less than four degrees. As the slope increases, NDVI in Shandong Province also tends to increase. At a slope of 18 degrees, NDVI stabilizes. Beyond a slope of 22 degrees, NDVI values become more variable. According to Figure 12c, NDVI

increases with increasing digital elevation model (DEM) values from 0 to 42 m; however, as DEM ranges from 40 to 60 m, NDVI decreases with increasing elevation; above 80 m, the relationship between NDVI and DEM is not apparent.

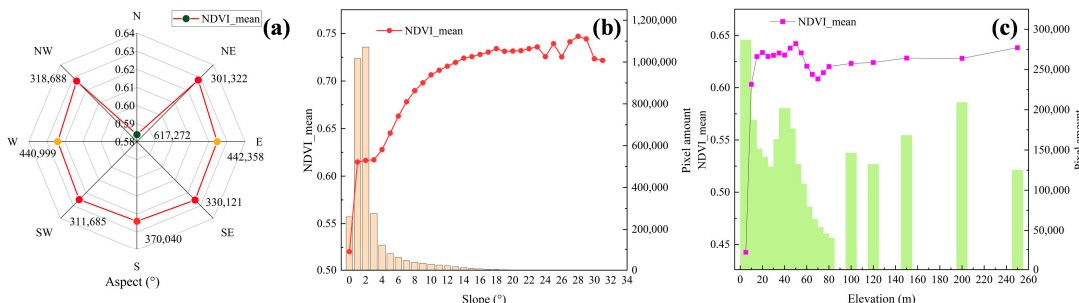

**Figure 12.** Topographic influences on normalized difference vegetation index (NDVI) in Shandong Province, utilizing digital elevation model (DEM): (**a**) aspect, (**b**) slope, and (**c**) elevation.

3.4.3. Comprehensive Response Analysis of NDVI to Climate, Topography, and Human Activities in Shandong Province

In order to probe the impact of various geographical factors on NDVI dynamics in Shandong Province, a total of 12 pertinent variables were selected for investigation across 39,391 sampling points in the province through Geodetector detection.

1. Data Processing for Geodetector

In this study, a Geodetector was employed to analyze twelve factors influencing the NDVI in Shandong Province. Prior to utilizing the Geodetector, it was necessary to classify the data. This classification was facilitated using R programming, applying five different methods: "equal" (equal interval method), "natural" (natural breaks), "quantile" (quantile method), "geometric" (geometric progression), and "sd" (standard deviation method). The outcomes of these classifications are detailed in Table 3. LAND and Soil, being inherently categorical variables, did not undergo further processing. For Aspect, which was divided into eight categories based on compass directions, the "natural" method was selected for classification.

2. Factor Detector

Spatiotemporal analysis has revealed that NDVI in Shandong Province exhibits significant spatial heterogeneity. However, the presence of multicollinearity among the explanatory variables could potentially hinder the model's ability to effectively interpret this heterogeneity. On the other hand, the Geodetector used in this study demonstrates robustness against the effects of multicollinearity among the independent variables, enabling the accurate assessment of individual factors' impacts on the spatial distribution of vegetation cover. The higher the $q$-value obtained from the factor detector, the greater the contribution of the factor to the response variable. Furthermore, the factor with the highest $q$-value is defined as the dominant factor. As shown in Table 4, although all factors significantly influenced the spatial variability of NDVI in the study years of 2005, 2010, 2015, and 2020 ($p < 0.01$), the extent to which each factor explained this variability varied. Ordered by explanatory power, these factors are Land, Soil, Light, Elevation, Road, Population, Temperature, Slope, River, PAR, PRE, and Aspect. Among them, Land had the highest $p$-value, reaching 0.313 in 2020, and explained more than 25% of the variability, making it the primary influencing factor on the spatial heterogeneity of NDVI in Shandong Province. This dominance is likely related to the impacts of human activities on land use and ecosystem dynamics.

As indicated in Figure 13, the $q$-values for Aspect, River, and Population (POP) remain consistently low with minimal variation, suggesting that their impact on NDVI changes is relatively minor.

**Table 3.** Number of breakpoints and methods of continuous variables to classification variables.

| Year | 2020 | | 2015 | | 2010 | | 2005 | |
|------|------|------|------|------|------|------|------|------|
| Factors | Methods | Intervals Num | Methods | Intervals Num | Methods | Intervals Num | Methods | Intervals Num |
| PRE | geometric | 10 | sd | 10 | equal | 9 | geometric | 10 |
| TEM | natural | 9 | quantile | 9 | quantile | 10 | natural | 10 |
| PAR | natural | 10 | natural | 8 | equal | 10 | equal | 10 |
| Elevation | geometric | 10 | geometric | 9 | geometric | 10 | geometric | 10 |
| Slope | sd | 10 | geometric | 9 | geometric | 10 | geometric | 9 |
| Aspect | natural | 8 | natural | 8 | natural | 8 | natural | 8 |
| Soil | | 22 | | 22 | | 22 | | 22 |
| LAND | | 5 | | 5 | | 5 | | 5 |
| POP | natural | 10 | natural | 10 | natural | 10 | natural | 10 |
| Light | geometric | 10 | natural | 10 | geometric | 10 | natural | 10 |
| River | equal | 10 | geometric | 8 | equal | 9 | geometric | 10 |
| Road | quantile | 10 | equal | 7 | equal | 7 | equal | 7 |

'PRE' stands for precipitation, 'TEM' for temperature, 'PAR' for photosynthetically active radiation, 'LAND' for land use type, and 'POP' for population density. Classification methods include 'geometric' (dividing data based on geometric progression), 'natural' (using natural breaks based on data distribution), 'sd' (standard deviation intervals), 'equal' (creating equal intervals), and 'quantile' (distributing data based on quantiles).

**Table 4.** Drivers of normalized difference vegetation index (NDVI) *q*-values in Shandong Province in 2005, 2010, 2015, and 2020.

| Year | Factors | PRE | TEM | PAR | Elevation | Slope | Aspect | Soil | LAND | POP | Light | River | Road |
|------|---------|-----|-----|-----|-----------|-------|--------|------|------|-----|-------|-------|------|
| 2005 | *q*-value | 0.021 | 0.021 | 0.020 | 0.103 | 0.030 | 0.011 | 0.214 | 0.300 | 0.025 | 0.172 | 0.010 | 0.017 |
| | sig | 0 | 0 | 0 | 0 | 0 | 0 | 0 | 0 | 0 | 0 | 0 | 0 |
| 2010 | *q*-value | 0.016 | 0.027 | 0.042 | 0.086 | 0.025 | 0.009 | 0.178 | 0.288 | 0.030 | 0.231 | 0.008 | 0.013 |
| | sig | 0 | 0 | 0 | 0 | 0 | 0 | 0 | 0 | 0 | 0 | 0 | 0 |
| 2015 | *q*-value | 0.019 | 0.032 | 0.013 | 0.109 | 0.017 | 0.005 | 0.162 | 0.262 | 0.033 | 0.212 | 0.008 | 0.014 |
| | sig | 0 | 0 | 0 | 0 | 0 | 0 | 0 | 0 | 0 | 0 | 0 | 0 |
| 2020 | *q*-value | 0.008 | 0.028 | 0.014 | 0.158 | 0.026 | 0.007 | 0.206 | 0.313 | 0.038 | 0.197 | 0.023 | 0.055 |
| | sig | 0 | 0 | 0 | 0 | 0 | 0 | 0 | 0 | 0 | 0 | 0 | 0 |

'PRE' stands for precipitation, 'TEM' for temperature, 'PAR' for photosynthetically active radiation, 'LAND' for land use type, and 'POP' for population density.

3.    Interaction Detector

In the longitudinal analysis of the relationship between the NDVI and key driving factors in Shandong Province, it was found that changes in NDVI are influenced not only by individual factors but also exhibit more complex dynamics when multiple factors interact. In particular, by comparing single-factor analysis and interaction detection, a deeper understanding of how these factors collectively affect vegetation cover was achieved.

In the single-factor analysis, significant effects of land use type (LAND), light exposure (LIGHT), and soil type on NDVI were observed. Changes in land use, such as the transformation of forested areas into agricultural lands or urban territories, directly altered the type and density of surface vegetation, subsequently affecting the NDVI values. Light exposure, a critical factor for plant growth, also had significant impacts on NDVI across different years.

However, through interaction detection, it was found that, when two or more factors acted simultaneously, their impact on NDVI exhibited patterns of bivariate enhancement and nonlinear strengthening, often exceeding the effects of individual factors. As shown in Figure 14, the dual interaction between land use and elevation in the analysis of 2020 demonstrated significant explanatory power (*q*-value of 0.4), indicating substantial differences in the impact of these combined factors on the spatial distribution of NDVI. Similarly, interactions involving light exposure and other factors also demonstrated variable trends, although their explanatory power fluctuated across different years.

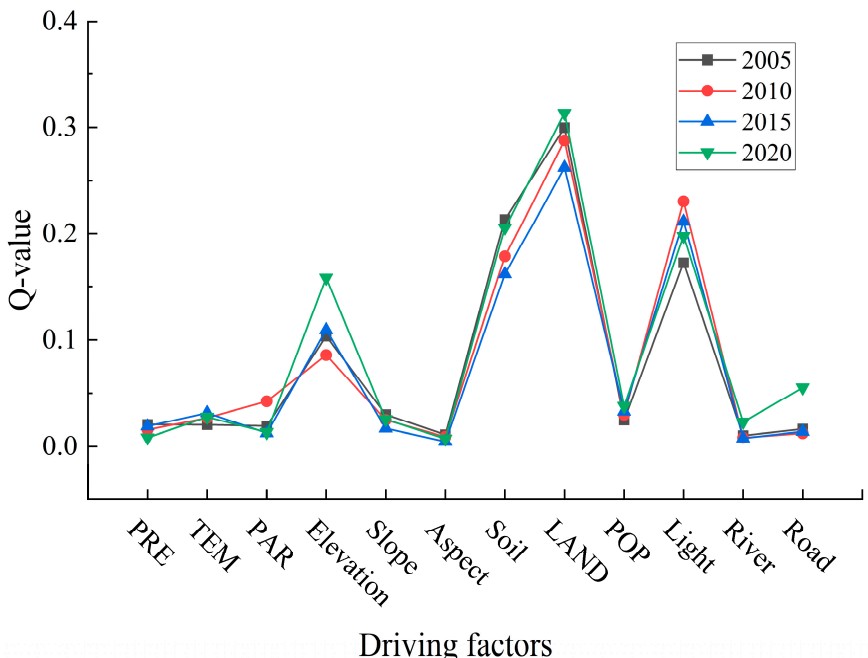

**Figure 13.** *Q*-values of various driving forces in 2005, 2010, 2015, and 2020. 'PRE' stands for precipitation, 'TEM' for temperature, 'PAR' for photosynthetically active radiation, 'LAND' for land use type, and 'POP' for population density.

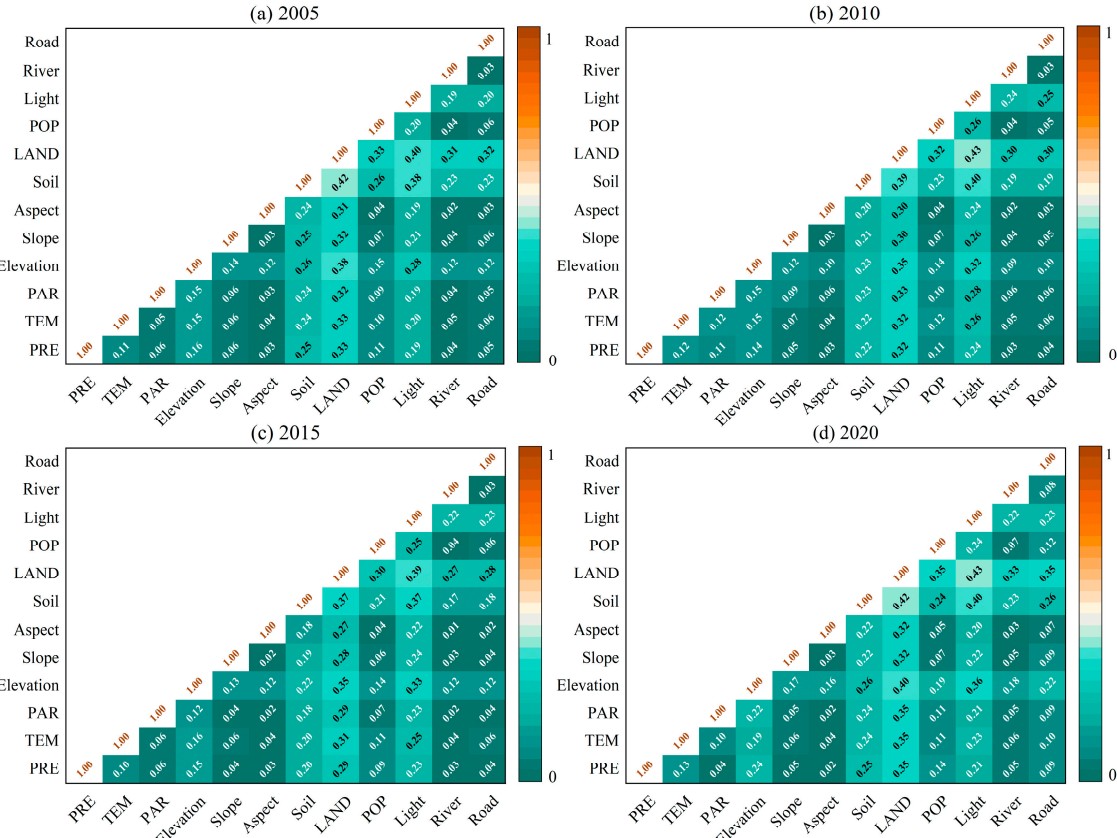

**Figure 14.** Yearly interaction detection of normalized difference vegetation index (NDVI) influencing factors in Shandong Province for (**a**) 2005, (**b**) 2010, (**c**) 2015, and (**d**) 2020. 'PRE' stands for precipitation, 'TEM' for temperature, 'PAR' for photosynthetically active radiation, 'LAND' for land use type, and 'POP' for population density.

The comprehensive analysis indicates that NDVI changes are influenced by a variety of factors, with interactions playing a particularly crucial role. Data from 2005 to 2020 indicate that land type and light exposure are primary influencing factors, though the extent of their impact varies over time. Furthermore, interaction analysis provides a more complex yet comprehensive method to understand these dynamics, particularly when assessing the long-term effects of environmental changes on ecosystems.

## 4. Discussion

### 4.1. Trends in Vegetation by NDVI Changes

This study employs the NDVI to examine the trends in vegetation cover in Shandong Province from 2001 to 2020. The findings indicate that vegetation cover in Shandong Province has exhibited complex spatiotemporal variations over the past two decades, with significant growth trends reflecting the success of regional vegetation restoration and ecological rehabilitation efforts. Compared to traditional studies, such as the analysis by Dong et al. [52], this research provides a more detailed description of spatiotemporal dynamics, revealing variations in the intensity of vegetation restoration across different regions. These variations may be related to the implementation intensity of regional ecological policies and community participation, particularly in ecological reserves and peri-urban areas where vegetation increases have been more significant.

### 4.2. The Relationship between Climatic Factors and Vegetation NDVI

According to the analysis in this study, there is a significant positive correlation between NDVI and both precipitation and temperature in Shandong Province, supporting the findings of Dong et al. [52] that climatic factors are key drivers of vegetation growth. Moreover, our research indicates that PAR has a complex effect on NDVI, suggesting that the role of sunlight may vary across different ecological zones due to limitations in moisture conditions. These findings emphasize the importance of how changes in precipitation and temperature, under the backdrop of global warming, can affect vegetation growth cycles and biomass accumulation, thereby impacting regional vegetation cover. Therefore, future models on the impact of climate factors on vegetation should consider these regional differences more thoroughly.

### 4.3. The Impact of Human Activities on Vegetation NDVI

Similarly to other studies [37,50,53], this research finds that human activities, particularly changes in land use, have a significant impact on NDVI. Through Geodetector analysis, we further confirm that changes in land cover type are a major human-driven factor affecting NDVI, especially in regions with intense agricultural activity and rapid urban expansion. Specifically, activities such as agricultural expansion, urbanization processes, and deforestation directly alter the coverage and structure of natural vegetation, impacting not only ecological functions but also potentially leading to biodiversity loss and land degradation. Hence, effective land management and vegetation restoration strategies are crucial for maintaining ecological balance and promoting sustainable development.

### 4.4. Limitations of the Study

Despite providing significant insights into the changes in vegetation cover in Shandong Province, this study has limitations. For example, the spatial resolution of the NDVI data used may not capture fine-scale vegetation dynamics adequately. Additionally, although meteorological data like photosynthetically active radiation are useful in explaining vegetation changes, limitations in the temporal coverage and resolution of the datasets might not fully reflect actual surface conditions. Consequently, there is an increasing need for high-precision and high-resolution remote sensing data to more accurately assess the impacts of climatic factors and human activities on vegetation dynamics.

*4.5. Future Research Directions*

In light of the findings and limitations of the current study, future research should consider the following directions: Firstly, introduce higher resolution and longer time series remote sensing data to enhance the precision and reliability of vegetation change monitoring. Secondly, employ more complex statistical models and machine learning techniques to analyze the nonlinear relationships and interactions between vegetation and climatic factors, as well as human activities. Furthermore, future research should consider biotic interactions and the time-lagged effects of abiotic factors. For example, climate change has a significant impact on seedling survival at the community level [39]. Additionally, a significant rainfall event might have delayed effects on vegetation growth, becoming evident only weeks or months later [40]. These factors are often subtle and complex, yet they can greatly enhance our understanding of the spatiotemporal variations in vegetation cover. Investigating these broader ecological variables can provide deeper insights into the relationship between vegetation dynamics and climate.

## 5. Conclusions

This study used NDVI as an indicator to investigate the spatiotemporal changes and driving forces of vegetation in Shandong Province from 2001 to 2020, employing trend analysis, partial correlation analysis, and the Geodetector model.

Throughout the study period in Shandong Province, NDVI data indicated signs of vegetation recovery, exhibiting an upward trend with a fluctuation rate of 0.0017. Initial stability was observed in the majority of areas, with more than 70% stability in 2001–2005 and over 45% in 2006–2010. A notable transition to increased vegetation health was evident in the later periods of 2011–2015 and 2016–2020. Partial correlation analysis showed NDVI to generally have a positive correlation with precipitation (average coefficient of 0.203) and a significant positive correlation with temperature (average of 0.283), but a predominantly negative correlation with photosynthetically active radiation (average of −0.159). Geodetector analysis identified land use type as the most influential factor on vegetation changes, followed by nighttime lighting, soil type, and elevation. The interaction between land use type and soil type was particularly significant, explaining 42% of the NDVI variation and highlighting the dominant influence of human activities on vegetation dynamics in the province.

The findings of this study offer vital insights for the formulation and decision-making processes concerning the ecological environment of Shandong Province, significantly contributing to the maintenance of ecological security and sustainable development in the region. In future work, we plan to incorporate diverse vegetation change indices, such as net primary production (NPP) and remote sensing ecological indexes. Additionally, we aim to examine the spatiotemporal variations in vegetation across Shandong Province by considering a broader range of variables, including those related to air quality, to provide a more comprehensive analysis.

**Author Contributions:** Conceptualization, D.D. and H.D.; methodology, D.G. and D.D.; software, D.D. and Y.Z.; formal analysis, Z.Z. and H.G.; data curation, D.D., Z.Z., H.G. and Y.Z.; writing—original draft preparation, D.D.; writing—review and editing, Y.F.; visualization, D.D., D.G. and H.D.; supervision, Y.F.; project administration, Y.F.; funding acquisition, Y.F. All authors have read and agreed to the published version of the manuscript.

**Funding:** This research was funded by the JST SPRING (grant number JPMJSP2136), JSPS KAKENHI (grant number JP21H05179), and the National Natural Science Foundation of China (grant numbers 32001315, U1809208, and 31870618).

**Data Availability Statement:** Data will be made available on request.

**Acknowledgments:** We appreciate the constructive suggestions and comments from the editor and anonymous reviewers.

**Conflicts of Interest:** The authors declare that they have no known competing financial interest or personal relationships that could have appeared to influence the work reported in this paper.

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
