# Peer review of "Analysis of Spatiotemporal Evolution and Driving Forces of Vegetation from 2001 to 2020: A Case Study of Shandong Province, China"

_forests, doi:10.3390/f15071245_

Round 1

Reviewer 1 Report

Comments and Suggestions for Authors

The manuscript has significant value in conservation science. The manuscript has several figures, some figure's content is overlapping with the table's content. Need to consider this issue. Other minor comments are given below:

Line 53: NDVI--- add full form for the first time.

Line 57: RWI--- add full form for the first time.

Line 142: twelve driving factors----- Whats are the twelve factors?

Line 176: Figure 2:---- Add a clear ligand for the figure.

Table 3: PRE, TEM, PER, POP---- mention full form in the caption of the Table.

Line 555-580: Conclusion--- need to rewrite compactly.

Reference list---- Check the abbreviations of the journal's names.

Reviewer 2 Report

Comments and Suggestions for Authors

This manuscript examined the spatiotemporal variation and potential driving factors of vegetation cover in Shandong Province in China from 2001 to 2020 using trend analysis methods, partial correlation analysis, and Geodetector based on the normalized difference vegetation index (NDVI) data combined with climatic, topographic, and anthropogenic activity data.  The paper is interesting and potentially useful, as it demonstrated an overall upward trend in vegetation cover in Shandong Province, particularly in areas with concentrated human activities, and made an attempt to suggest that climatic factors exhibited a positive correlation with vegetation growth, whereas land use changes emerged as one of the key drivers influencing vegetation dynamics in the study area.  I think that the combined use of NDVI data with climatic, topographic, and anthropogenic activity data in a trend analysis, partial correlation analysis, and Geodetector modeling framework is an interesting advantage of this paper in relation to others that deal with similar topics.  There are some justifications the authors need to include, which will enrich the content of the research while clarifying the selection and implementation of the approaches used.  The specific comments are:

Abstract and Keywords

-        (1)  Page 1 Line 33.  “NDVI” to “Normalized Difference Vegetation Index (NDVI)”.  In the Keywords section, I would suggest specifying the full name of NDVI, because people who are not in this field may not be quite familiar with this abbreviation.

Introduction

-        (2)  Page 2 Line 53.  You may want to specify here what NDVI represents, as this is the first time it appears in the main text.

-        (3)  Page 2 Line 57.  What does “RWI” stand for?  Please specify it here.

-        (4)  Page 2 Line 76.  “in-appropriate” to “inappropriate”.

-        (5)  Page 2 Line 79.  “methods commonly used” to “methods that are commonly used”.

-        (6)  Some additional information, such as brief background on some available international ecological studies on the dynamics of vegetation and related potential driving factors, would be helpful in the Introduction section to set the stage.  I would also suggest emphasizing somewhere in the Introduction section the significance of the combined use of NDVI data with climatic, topographic, and anthropogenic activity data in a trend analysis, partial correlation analysis, and Geodetector modeling framework, because these approaches might help people from broader field find your work useful rather than people just working on the particular variable or area being interested in it.

Materials and Methods

-        (7)  Page 3 Line 121.  “warm temperate” to “warm-temperate”.

-        (8)  Page 3 Lines 124-125.  “warm temperate” to “warm-temperate”.

-        (9)  Page 4 Line 136.  “human activity data” to “human-activity data”.

-        (10)  Page 4 Line 143.  “twelve driving factors” to “twelve potential driving factors”.

-        (11)  Page 4 Line 153.  “from CLCD” to “from the China Land Cover Dataset (CLCD)”.  I would suggest specifying here what CLCD represents, because this is the first time it appears in the main text and people who are not in this field may not be quite familiar with the abbreviation.

-        (12)  Page 4 Line 160.  “from NOAA” to “from the United States National Oceanic and Atmospheric Administration (NOAA)”.  Please specify here the full name of NOAA, as this is the first time it appears in the main text.

-        (13)  Page 4 Lines 135-168.  Since the climatic, topographic, and anthropogenic activity data might be produced by different remote-sensing technologies and have different resolutions, I would suggest adding a bit more description of how you matched the resolution for the variables you extracted from different data sources in this study.

-        (14)  Page 4 Table 1 Line 169.  I would suggest specifying in the table caption what “16d”, “1a”, and “GEE” stand for, because people who are not in this field may not be quite familiar with these abbreviations.

-        (15)  Page 4 Table 1.  “Data set” to “Dataset”.

-        (16)  Page 5 Figure 2 Line 177.  I would suggest specifying in the figure caption what “NDVI”, “PRE”, “TEM”, “PAR”, “LAND”, and “POP” represent.  A good figure or table caption should make the figure or table understandable without reference to the main text.

-        (17)  Page 6 Line 185.  “is employed” to “are employed”.

-        (18)  Page 7 Lines 219-223.  There needs some further explanation as to why partial correlation analysis was chosen in this study.  There are some other statistical approaches that have been used in ecology when it comes to analyzing the relationships between variables.  However, there is limited justification as to why the authors only chose partial correlation analysis.  I would suggest adding a brief explanation to justify why this approach and not other also commonly used ones was chosen.  Given the potential of statistical approaches to influence the assessment of relationships between variables, it would be better to critically assess why some approaches are chosen and their potential and weaknesses for the available data and specific environment considered.

-        (19)  Page 7 Lines 225-237.  When you used partial correlation analysis to examine the relationships between variables in this study, did you use raw or transformed values of the variables to apply partial correlation analysis on?  It would be better to specify a little more even though data transformation may not be necessary for some situations, because it is an important step to make sure the variables meet the underlying assumptions of the algorithms before conducting any statistical analyses.  Statistical approaches with different mechanisms may have different underlying assumptions of normality, linearity or multicollinearity, and some variables may need to be transformed to meet specific assumptions.

Results

-        (20)  Page 8 Figure 3 Line 284.  “NDVIage” to “Normalized Difference Vegetation Index (NDVI)”.

-        (21)  Page 9 Figure 4 Line 287.  “monthly mean NDVI” to “Monthly mean Normalized Difference Vegetation Index (NDVI)”.

-        (22)  Page 9 Figure 5 Line 290.  “NDVI” to “Normalized Difference Vegetation Index (NDVI)”.

-        (23)  Page 11 Figure 6 Line 331.  “NDVI Value Distribution by Area and Percentage (b) and” to “Normalized Difference Vegetation Index (NDVI) value distribution by area and percentage (b), and”.

-        (24)  Page 11 Figure 7 Line 341.  “NDVI” to “Normalized Difference Vegetation Index (NDVI)”.

-        (25)  Page 12 Figure 8 Line 352.  “NDVI” to “Normalized Difference Vegetation Index (NDVI)”.

-        (26)  Page 12 Figure 8 Lines 352-354.  “Green is a significantly improved area, light green is a slightly improved area, yellow is a stable area, and orange is a slightly degraded area. Red indicates severely degraded areas” to “Green indicates significantly improved area, light green indicates slightly improved area, yellow indicates stable area, orange indicates slightly degraded area, and red indicates severely degraded area”.

-        (27)  Page 12 Figure 9 Line 356.  “NDVI” to “Normalized Difference Vegetation Index (NDVI)”.

-        (28)  Page 12 Figure 10 Line 358.  “NDVI” to “Normalized Difference Vegetation Index (NDVI)”.

-        (29)  Page 12 Figure 10 Line 360.  “and red” to “and red indicates severely degraded area ratio”.

-        (30)  Page 13 Figure 11 Line 380.  “(a) annual” to “(a), annual”.

-        (31)  Page 13 Figure 11 Line 380.  “(b) and” to “(b), and”.

-        (32)  Page 13 Figure 11 Line 381.  I would suggest removing “distributed”.

-        (33)  Page 13 Figure 12 Line 383.  “NDVI” to “Normalized Difference Vegetation Index (NDVI)”.

-        (34)  Page 13 Figure 12 Line 384.  “(a), and annual” to “(a), annual”.

-        (35)  Page 13 Figure 12 Line 384.  It seems the description of panel (c) is missing.

-        (36)  Page 13 Figure 12 Line 384.  “in China” to “in Shandong Province”.

-        (37)  Page 14 Figure 13 Line 413.  “NDVI” to “Normalized Difference Vegetation Index (NDVI)”.

-        (38)  Page 14 Figure 13 Line 413.  “aspect, slope and elevation” to “aspect, slope, and elevation”.

-        (39)  Page 14 Figure 13 Line 413.  I would suggest specifying in the figure caption what the panels (a), (b), and (c) represent respectively.

-        (40)  Page 14 Figure 13 Line 413.  Please specify in the figure caption what the “DEM” in this figure stands for.

-        (41)  Page 14 Line 418.  I would suggest using “four” rather than “4” when the number is less than ten.

-        (42)  Page 15 Table 2 Line 439.  I would suggest specifying in the table caption what “PRE”, “TEM”, “PAR”, “LAND”, “POP”, “geometric”, “natural”, “sd”, “equal”, and “quantile” represent.

-        (43)  Page 16 Table 3 Line 463.  “NDVI” to “Normalized Difference Vegetation Index (NDVI)”.

-        (44)  Page 16 Table 3 Line 463.  “2010, 2015 and 2020” to “2010, 2015, and 2020”.

-        (45)  Page 16 Table 3 Line 463.  Please specify in the table caption what “PRE”, “TEM”, “PAR”, “LAND”, and “POP” stand for.

-        (46)  Page 16 Figure 14 Line 466.  “q value” to “q-value”.

-        (47)  Page 16 Figure 14 Line 466.  “2010, 2015 and 2020” to “2010, 2015, and 2020”.

-        (48)  Page 16 Figure 14 Line 466.  Please specify in the figure caption what “PRE”, “TEM”, “PAR”, “LAND”, and “POP” represent.

-        (49)  Page 17 Line 485.  “q value” to “q-value”.

-        (50)  Page 17 Figure 15 Line 490.  “NDVI” to “Normalized Difference Vegetation Index (NDVI)”.

-        (51)  Page 17 Figure 15 Line 490.  I would suggest specifying in the figure caption what the panels (a), (b), (c), and (d) represent respectively.

-        (52)  Page 17 Figure 15 Line 490.  Please specify in the figure caption what “PRE”, “TEM”, “PAR”, “LAND”, and “POP” stand for.

Discussion and Conclusions

-        (53)  Page 18 Line 537.  “data sets” to “datasets”.

-        (54)  Besides the climatic, topographic, and anthropogenic activity variables examined in this study, the spatiotemporal variation of vegetation cover in Shandong Province may also be affected by potential factors such as biotic interactions and time-lagged abiotic conditions on a finer temporal scale (e.g., one-month prior precipitation and temperature).  I would suggest briefly acknowledging in the Discussion section these additional factors based on literature and their potential influence on the spatiotemporal variation of vegetation cover in the study area.

Comments on the Quality of English Language

I would suggest minor editing of English language.

Reviewer 3 Report

Comments and Suggestions for Authors

The manuscript entitled “Analysis of Spatiotemporal Evolution and Driving Forces of Vegetation from 2001 to 2020: A Case Study of Shandong Province, China” presents the spatio-temporal variation of vegetation via the NDVI and links this variation with different governing factors.

Major comments

-          The novel of the study has not been clarified in the introduction

-          The methodology was not clear. Why authors used different methods (partial correlation analysis, and Geodetector) to investigate the relationship between the spatio-temporal variation of vegetation with driving factors? For analyzing this relationship, it is better to separate the factors that control the temporal variations (e.g., precipitation, temperature) with the factors that control spatial variation. For long-term temporal variation, it is more useful if the analysis is performed for the wet and dry seasons. In addition, the reason for temporal variation was not clear.

-          The result of trend analysis using Mann-Kendall trend test was not presented. The trend is significant or not?

Minor comments

Introduction: This study used methods developed by other previous studies to explore the spatiotemporal evolution of vegetation in Shangdong. However, previous studies related to this paper in the province were not mentioned. Why studying the vegetation dynamics in Shangdong is important? Please provide the novel of this study compared to the previous studies.

Line 134-168: Please re-write these paragraphs to clarify the data you used

Figure 2: Check the caption

Line 255: Correct the explanations

Lines 325-329: I don’t understand the explanations for Figure 6c. The y-axis of Figure 6b,c is not the same scale so I can’t interpret it.

Lines 332-339: What are criteria for high to extremely low of the NDVI? What is the message of this paragraph and Figure 7?

Lines 340-350: Again, what are criteria for “significantly improved area, slightly improved area, a stable area, slightly degraded area and severely degraded area? Also, please explain the reason for this temporal variation.

Line 361-373: Please describe correctly Figure 10. For example, figure 10 shows that the 2001-2005, a large portion of the Shandong is slight improvement (not stable as described in the text).

Line 384: in China?

Line 508: What do you mean “climate change”?

Line 520: The only factor population density can represent human activities?

Comments on the Quality of English Language

quality of English is acceptable

Round 2

Reviewer 2 Report

Comments and Suggestions for Authors

The authors have addressed most of the comments.  I just have some minor suggestions below, which I hope could help further improve the clarity of the paper.

For the revised version:

-        (1)  Page 1 Lines 34-35.  Please arrange the keywords alphabetically.

-        (2)  Page 2 Line 59.  “(RWI)time” to “(RWI) time”.

-        (3)  Page 4 Figure 1 Line 156.  “area, (c)” to “area, and (c)”.

-        (4)  Page 5 Line 192.  “sourced from from” to “sourced from”.  Please remove the duplicated “from”.

-        (5)  Page 7 Figure 2 Lines 218-220.  I would suggest specifying in the figure caption the full name of “SEN Trend analysis”, because people who are not in this field may not be quite familiar with this abbreviation.  A good figure or table caption should make the figure or table understandable without reference to the main text.

-        (6)  Page 9 Line 277.  “Where” to “where”.

-        (7)  Page 17 Figure 11 Lines 445-446.  “and solar radiation (PAR)” to “and photosynthetically active radiation (PAR)”.

-        (8)  Page 17 Figure 12 Line 449.  “annual annual mean temperature” to “annual mean temperature”.  Please remove the duplicated “annual”.

-        (9)  Page 18 Figure 13 Line 477.  It seems the x axis of panel (c) should be “Elevation (m)” rather than “DEM (m)”.

-        (10)  Page 18 Figure 13 Line 479.  “Slope, (c)” to “Slope, and (c)”.

-        (11)  Page 20 Figure 14.  I would suggest arranging the legend in this figure from 2005 to 2020 rather than from 2020 to 2005.

-        (12)  Page 20 Figure 14 Lines 540-541.  It seems “Interaction Detector and Ecological Detector” should be put in the main text as the next section.

-        (13)  Page 21 Figure 15 Line 564.  “, (d)” to “, and (d)”.

-        (14)  Page 21 Figure 15 Line 564.  The years of “(a) 2005, (b) 2010, (c) 2015, (d) 2020” in the figure cation are different from the years in the figure.  Please check which one is correct and make them consistent.

Reviewer 3 Report

Comments and Suggestions for Authors

Comments on the Quality of English Language
